# Resveratrol in Liquor Exacerbates Alcoholic Liver Injury with a Reduced Therapeutic Effect in Mice: An Unsupervised Herbal Wine Habit Is Risky

**DOI:** 10.3390/nu14224752

**Published:** 2022-11-10

**Authors:** Songxia Zhang, Ying Xu, Mengling Ye, Wenli Ye, Jian Xiao, Honghao Zhou, Wei Zhang, Yan Shu, Yun Huang, Yao Chen

**Affiliations:** 1Department of Clinical Pharmacology, Xiangya Hospital, Central South University, Changsha 410008, China; 2Institute of Clinical Pharmacology, Central South University, Changsha 410078, China; 3Engineering Research Center of Applied Technology of Pharmacogenomics, Ministry of Education, Changsha 410078, China; 4National Clinical Research Center for Geriatric Disorders, Changsha 410008, China; 5Department of Pharmacy, Xiangya Hospital, Central South University, Changsha 410008, China; 6Department of Pharmaceutical Sciences, School of Pharmacy, University of Maryland, Baltimore, MD 21201, USA; 7Department of Hepatobiliary Surgery, Xiangya Hospital, Central South University, Changsha 410008, China

**Keywords:** herbal wine use, resveratrol, ethanol, concomitant administration, liver injury

## Abstract

People in Eastern countries hold a tradition of soaking herbal medicine in wine; however, the efficacy and safety of herbal wine have not been rigorously assessed. By assessing the efficacy of resveratrol (RSV) in ethanol against alcoholic liver disease (ALD) in mice, we aimed to offer a perspective on the use of herbal wine. To simulate the behaviour of herbal wine users, RSV (15 mg/kg) soaked in ethanol (RSV-alcohol) was administrated via gavage to the mice, here with alcohol consumption-induced ALD. RSV soaked in water (RSV-water) was the treatment control. The efficacy and safety of RSV on ALD were evaluated. Compared with the RSV-water group, a higher rate of mortality was found in the RSV-alcohol group (50.0% vs. 20.0%), which also exhibited more severe liver injury. RSV significantly increased the exposure of alcohol by 126.0%, which was accompanied by a significant inhibition of the ethanol metabolic pathway. In contrast, alcohol consumption significantly reduced exposure to RSV by 95.0%. Alcohol consumption had little effect on the expression of drug-metabolizing enzymes in RSV; however, alcohol seemed to reduce the absorption of RSV. RSV in liquor exacerbates alcoholic liver injury and has a reduced therapeutic effect, suggesting that the habit of herbal wine use without supervision is risky.

## 1. Introduction

Alcohol consumption has a serious impact on human health, and its abuse leads to alcoholic liver disease (ALD) [1]. ALD presents as a range of disorders of different severity, including simple steatosis or fatty liver, alcoholic hepatitis, fibrosis, or cirrhosis. Worldwide, its incidence increases year by year [2,3]. In 2016, 5.1% of the global disease burden was caused by alcoholic beverage consumption, and alcohol-attributable liver cirrhosis caused 607,000 deaths and 22.2 million disability-adjusted life years (DALYs) [4]. In 2018, alcohol consumption caused 3 million deaths and 132.6 million DALYs [5]. In 2020, an estimated 4.1% of all new cases of cancer were attributable to alcohol consumption [6]. Regarding China, it has been reported that hospital admissions of alcoholic cirrhosis increased by 33% among the 2.3 million inpatients in Beijing’s 31 top-ranking hospitals [7].

The treatment of ALD remains challenging. The current treatment options mainly include immediate abstinence from alcohol, nutritional support, and medication [1]. Clinically, corticosteroids, pentoxifylline, and antioxidants, such as vitamin E and *N*-acetylcysteine are the most commonly used drugs in treating ALD [8]. With accumulating evidence of their efficacy and low toxicity, natural compounds such as resveratrol (RSV) [9], flavonoids [10], saponins [11] and β-carotene [12] have begun to offer additional pharmacological options for treating ALD. Among these natural compounds, RSV is one of the most promising multifunctional drugs for ALD. 

RSV (3,5,4′-trihydrocxy-trans-stilbene) is a nonflavonoid phenol produced by numerous plants in response to bacteria or fungi colonisation. RSV is mainly found in Polygonum cuspidatum, grape skin, peanuts, and berries [13]. RSV has been characterised as an anti-inflammatory, antioxidative, anticancer, and antiageing compound that can be used in the treatment of malignant tumours, neurodegenerative diseases, and cardiovascular diseases [14]. Possibly because of its anti-inflammatory, antioxidant, and calorie-restricting effects, RSV has shown promising efficacy against liver disorders [15]. For example, RSV can significantly increase the survival rate after liver transplantation and can reduce fat deposition, ischaemia-induced necrosis, and apoptosis in rats [15]. RSV can protect the liver from chemical, cholestatic, and alcohol-induced damage in rodents [16,17]. RSV can also improve glucose metabolism and lipid profiles while reducing reduce liver fibrosis and steatosis [18]. In particular, RSV has been shown to possess therapeutic effects in alcohol-induced liver damage [19]. 

RSV is an important component in traditional Chinese medicine (TCM), which has played a vital role in the treatment of diseases and health conditions for centuries in China [20,21]. Certain TCM is often used together with alcohol, known as herbal wine [22]. For a long time, people in Eastern countries have gradually formed the habit of using herbal wine to treat diseases, including ALD [23,24]. It is a popular belief that herbal wine can promote the curative effects of TCM, prolong life expectancy, and improve physical health. A variety of homemade herbal wines are widely consumed in daily Chinese life. However, there remains a lack of in-depth understanding of herbal wine use. The irrational and nonstandard consumption of herbal wine is common, which often leads to intoxication and even death [25]. Therefore, with the urgent need for scientific evidence to guide the rational use of herbal wine, the main purpose of the present study is to offer a perspective on the use of herbal wine by assessing the efficacy of RSV in ethanol against ALD in mice.

## 2. Materials and Methods

### 2.1. Chemicals and Reagents

Trans-resveratrol (purity: 99%, Lot No. E1711079) was purchased from Aladdin Industrial Corporation (Shanghai, China). Trans-resveratrol 3-o-β-d-glucuronide (purity: 97.23%, Lot No. 6-LXS-20-2) and trans-resveratrol-3-sulfate sodium salt (purity: 96.76%, Lot No. 10-UPA -21-2) were purchased from Toronto Research Chemicals (Brisbane Road, Toronto, ON, Canada). Glass capillaries (40 μL, coated with heparin sodium) were purchased from the Zibo Laixu Medical Equipment Co., Ltd., (Zibo, China). Mice restraints were purchased from Zhongke Life Science & Technology Co., Ltd., (Hangzhou, China). All other reagents and supplies were commercially available.

### 2.2. RSV Treatment in ALD Mice

C57BL/6 male mice aged 8–10 weeks were purchased from Hunan Slikejingda Experiment Animal Co., Ltd., (Changsha, China). The mice were housed in the Facility of Laboratory Animals of Central South University with a free case of water and food. The study was approved by the Ethics Committee of the Institute of Clinical Pharmacology, Central South University (Changsha, China). All the operations complied with the ethics and regulations related to animal experiments and strictly complied with the relevant regulations of the Administration Committee of Experimental Animals of Central South University (No: 2018sydw0219, Date: 18 October 2018, Changsha, China).

The mice were randomly divided into four treatment groups (n = 10 for each group): healthy control group (vehicle), ALD group (model), ALD treated with RSV by co-administration (RSV-alcohol), and ALD treated with RSV dissolved in water after a 6 h-interval (RSV-water). All treatments were administered by gavage. The ALD was established by gavage with alcohol for 2 weeks (30.3% of alcohol in the first week and 50.6% in the second week, both in volumes of 10 mL/kg), while the healthy control group was given the same volume of distilled water. To mimic the habit of herbal wine use, RSV was dissolved in alcohol directly used for ALD modelling and compared with that dissolved in water for 2 weeks. The concentration of RSV was 15 mg/mL. The intragastric volume of gavage for RSV-alcohol and RSV-water was 10 mL/kg, which was translated into a dose of 150 mg/kg RSV. 

### 2.3. Pharmacokinetics of RSV and Alcohol in Mice

To further understand the potential interaction between RSV and alcohol, we designed a pharmacokinetic study in mice. C57BL/6 mice were randomly divided into three groups (n = 6 for each group), including the RSV group, alcohol group, and RSV-alcohol group. All mice fasted for 12 h before the study. Then, a single dose of RSV in water (15 mg/kg), ethanol (45%, *v*/*v*), or RSV (15 mg/kg) in ethanol (45%, *v*/*v*) was dosed to the mice using an intragastric gavage. A series of blood samples at 0, 0.5, 0.75, 1, 2, 4, 6, 8 and 12 h (about 20 μL for each time point) from each mouse were collected through the tail vein by glass microcapillaries, as described previously [26]. Blood samples were stored in a −80 °C refrigerator away from light until analysis.

For determining RSV and its main metabolites, a Triple QuadTM 6500 HPLC-MS/MS system (AB Sciex, Concord, ON, Canada) equipped with an ACE Excel 5 Super C18 column (50 mm × 2.1 mm, 5 μm, ACE, Kent, WA, USA) was used. The mobile phase, which consisted of 5 mM ammonium acetate and acetonitrile at a flow rate of 0.30 mL/min was used for the separation of RSV, trans-resveratrol-3-o-β-d-glucuronide(R3G), trans-resveratrol-3-sulfate sodium salt (R3S) and diethylstilbestrol (internal standard, IS). The mouse blood sample (10 μL) was mixed with 200 μL methanol containing IS (5.48 ng/mL), which was vortexed and then centrifuged. The supernatant was transferred for HPLC-MS/MS detection. The mass spectrometer was operated in a negative ion mode through the use of multiple-reaction monitoring (MRM). The precursor–product ion transitions were monitored at *m*/*z* 226.9–184.9 for RSV, *m*/*z* 403.1–227.3 for R3G, *m*/*z* 309.0–227.1 for R3S, and *m*/*z* 267.4–237.3 for IS. Data acquisition was performed with the Analyst software (Version 1.4.2, Concord, Ontario, Canada).

For determining alcohol, a 7890GC-5975 GC-MS system equipped with an EI ion source (Agilent, Santa Clara, CA, USA) was used. A VF-WAXms Capillary column (30 m × 0.25 mm × 0.25 μm, Agilent, USA) was used to separate the analytes. Helium was used as the carrier gas. The temperature was set at 230 °C for the ion source, 150 °C for the four-stage rod, and 250 °C for the auxiliary heater, respectively. The programmed heating procedure for alcohol was as follows: initially, 40 °C for 3 min, which was increased from 20 °C/min to 90 °C for 2 min. Other instrument parameters included the following: the inlet temperature was 100 °C, the fractional flow rate was 20:1, the injection volume was 1 μL, and the flow rate was 0.50 mL/min. The mouse blood sample (10 μL) was mixed with 200 μL acetone containing 1 M *n*-butyl alcohol (IS), which was vortexed and centrifuged. The supernatant was transferred for GC-MS detection. A selective ion scan (SIM) mode was used for quantitative analysis. The quantitative ion was *m*/*z* 31.1 for alcohol and *m*/*z* 31.1 for IS. Data acquisition was performed using the NIST mass spectrometer library.

### 2.4. Cell Culture and Viability Assay

HL7702 cells were purchased from the Cell Bank of Type Culture Collection of the Chinese Academy of Sciences, Shanghai Institute of Cell Biology, Chinese Academy of Sciences. Cells were cultured in DMEM (Dulbecco’s modified Eagle medium) medium with 10% (*v*/*v*) foetal calf serum (GIBCO, Suzhou, China), 2.0 mM L-glutamine, 1.5 g/L NaHCO_3_, 0.1 mM nonessential amino acids, 1.0 mM sodium pyruvate, 100 IU/mL penicillin, and 100 μg/mL streptomycin. Cells were maintained in a 37 °C incubator (Thermo, Mariette, OH, USA) with 5% CO_2_ in humidified air. 

To investigate the cytotoxicity of alcohol, RSV, or alcohol combined with RSV, HL7702 cells were first plated in 12-well plates for 24 h and then treated with blank, 10 mM alcohol, 10 μM RSV or 10 mM alcohol together with 10 μM RSV for another 24 h. The cellular morphology was observed and photographed using an inverted light microscope. We then determined the IC50 of cell viability for these treatments in the HL7702 cells. HL7702 cells were plated in 96-well plates for 24 h and then treated with a series of concentrations for alcohol with and without 10 μM RSV and a series of concentrations for RSV with and without 30 mM alcohol. The cell viability was assessed by MTT assays, and IC50 was determined. 

### 2.5. Liver Function Analysis

Liver function was determined by the levels of serum biochemicals, including alanine aminotransferase (ALT), aspartate aminotransferase (AST), total protein (TP) albumin (ALB), cholesterol (CHOL), globulin (GLOB), triglyceride (TG), alkaline phosphatase (ALP), total cholesterol (TC), and high-density lipoprotein (HDL), with an automatic biochemical analyzer (Hunan Desheng Medical Equipment Co., Ltd., Changsha, China). 

### 2.6. Liver Histology

Small pieces of liver tissues were fixed in 4% paraformaldehyde, embedded in paraffin, cut into 5 μm-thick sections, and stained with H&E. The degree of liver steatosis was examined under an inverted microscope (Nikon ECLIPSE Ts2R; Nikon, Tokyo, Japan) and then photographed.

### 2.7. Enzymatic Activities of Adh, Aldh and Cyp2e1

A portion of the liver tissues was weighed and homogenised. The activities of Adh and Aldh were detected by an alcohol dehydrogenase activity assay kit and an Aldehyde dehydrogenase activity assay kit, respectively, according to the manufacturer’s instruction (Nanjing Jiancheng Institute of Biological Engineering, Nanjing, China). 

To determine the activity of Cyp2e1, liver microsomes were first prepared. In brief, the weighed liver tissues were put into a cold PBS buffer (pH = 7.4) at a ratio of 1:4 and homogenised. The homogenate was centrifuged at 4 °C by 10,000× *g* for 20 min. The supernatant was transferred and further centrifuged at 4 °C by 105,000× *g* for 60 min. The second supernatant was discarded, and the precipitated liver microsomes were resuspended in PBS buffer with a volume of 1 mL PBS per 1 g of initial liver tissues. The liver microsomes were stored at −80 °C for later use. Chlorazoxazone was used as the probe drug to determine the activity of Cyp2e1 according to our previous method [27,28]. The total incubation volume of the reaction was 200 μL, including 1 μL of probe drug solution (chlorzoxazone, 4.81 μg/mL) and 50 μL of liver microsome (final concentration 0.5 mg/mL). The reaction was initiated by adding 149 μL of NADPH reductive mixture and then incubated at 37 °C for 5 min. The reaction was stopped by adding 400 μL of cold acetonitrile containing 23.3 ng/mL glycyrrhetinic acid (IS), followed by centrifugation (4000 rpm) for 10 min at 4 °C. The supernatant was transferred and analysed by HPLC-MS/MS. The ratio of chlorazoxazone metabolite 6-hydroxy-chlorazoxazone to the parent drug resulting from the reaction was used to assess the activity of Cyp2e1.

A Triple QuadTM 6500 HPLC-MS/MS system (AB Sciex, Concord, ON, Canada) equipped with an ACE Excel 5 Super C18 column (50 mm × 2.1 mm, 5 μm, ACE, USA) was used to analyse chlorazoxazone and its metabolite 6-hydroxy-chlorazoxazone. A mobile phase consisting of 0.1% formic acid in water (A) and acetonitrile (B) set at a flow rate of 0.35 mL/min was used for separating chlorzoxazone, 6-hydroxy-chlorazoxazone, and IS. The gradient of the mobile phase was programmed as follows: 0–0.2 min, 10% of B; 0.2–3.5 min, 10–90% of B; 3.5–4.5 min, 90% of B; and 4.5–4.6 min, 90–10% of B. This was followed by re-equilibration at 10% of B for 5.0 min. The mass spectrometer was operated in a negative ion mode using MRM. The precursor–product ion transitions were monitored at *m/z* 168.1–132.0 for chlorzoxazone, *m/z* 184.1–120.1 for 6-hydroxy-chlorazoxazone and *m/z* 469.3–355.3 for IS. Data acquisition was performed using Analyst software, version 1.6.3 (AB Sciex, Concord, ON, Canada).

### 2.8. Immunoblotting

Mouse liver tissues were homogenised, and the lysate was prepared by using RIPA (Beyotime Institute of Biotechnology, Shanghai, China) according to the manufacturer’s protocol. The protein content was determined using a BCA protein assay reagent (Beyotime Institute of Biotechnology, Shanghai, China). Proteins (30 μg) were separated using 4–20% SDS-polyacrylamide gels, depending on the molecular weights of the target proteins, before being transferred to polyvinylidene fluoride membranes (Millipore, Bedford, MA, USA). The membrane was blocked with 5% nonfat milk for 2 h at room temperature. After incubation with primary antibodies at 4 °C overnight, the membrane was then incubated with horseradish peroxidase (HRP)–conjugated secondary antibodies (Proteintech, Wuhan, China) for 1 h at room temperature before being imaged using an Immobilon Chemiluminesent HRP substrate and a ChemiDoc TM XRS+ Gel Imaging System (BIO-RAD, CA, USA). The densities of the protein bands were analysed using Image Studio Lite (v. 5.2) software. Mouse monoclonal antibodies, including anti-Cyp2e1 (Abcam, Cambridge, UK), anti-Adh (Abcam, Cambridge, UK), anti-Aldh (Abcam, Cambridge, UK), anti-Sult1a1 (Bioworld, Visalia, CA, USA), anti-Ugt1a1 (Bioworld, USA), and anti-β-actin (Sigma Aldrich, St. Louis, MI, USA, GEA), were used as the primary antibodies. The relative expression of the target protein was reported as the optical density (OD), which was adjusted by the corresponding OD of β-actin.

### 2.9. Statistical Analysis

GraphPad Prism 8.0 (GraphPad Software, San Diego, CA, USA) was used for the graph generation. The pharmacokinetic parameters of RSV and alcohol were obtained using a noncompartmental analysis of the data with Drug and Statistical software, Version 3.2.2 (Clinical Drug Evaluation Centre, Wannan Medical College, Wuhu, China). The maximal plasma concentration (C_max_) and time to C_max_ (T_max_) were noted directly from the measured data. The elimination constant (k_e_) was calculated by the log-linear regression of the concentrations observed during the terminal phase of elimination, and the elimination half-life (t_1⁄2_) was then calculated as 0.693/k_e_. The area under the plasma concentration-time curve (AUC_(0–t)_) for the last measurable plasma concentration (C_t_) was calculated using the linear trapezoidal rule. The area under the plasma concentration–time curve for time infinity (AUC_(0–∞)_) was calculated as AUC_(0–t)_+C_t_ ⁄k_e_. Statistical analyses were performed using an unpaired Student’s *t*-test in SPSS 22 software (IBM Company, New York, NY, USA). A chi-square test was applied for nonparametric statistics, and significant differences were analysed using two-tailed tests (*p* < 0.05). 

## 3. Results

### 3.1. Development of ALD in Mice

As shown in Figure 1A, the C57BL/6 mice received a daily gavage of alcohol to establish ALD. Compared with the vehicle (distilled water) treatment, alcohol treatment had a significant impact on mouse physiology. Although all vehicle-treated mice survived, the survival rate was 80% for alcohol-treated mice (Figure 1B). The body weight of alcohol-treated mice was significantly decreased by 18.1% (Figure 1C), while its ratio of liver to body weight significantly increased by 13.6% (Figure 1D). The serum levels of ALT and AST in the alcohol-treated mice were significantly higher than those in the vehicle-treated mice. In contrast, the serum levels of TP and ALB were significantly lower in the former than in the latter (Figure 1E, Table 1). In particular, the H&E staining of liver tissues for the alcohol-treated mice showed apparent bullous steatosis, liver cell disorder, cell enlargement, cellular nucleus enlargement, and lipid droplets forming around the central vein (Figure 1F). Overall, the results indicated that our protocol of alcohol treatment protocol could successfully lead to ALD in mice.

### 3.2. RSV-Alcohol Treatment Worsened ALD in Mice

Compared with ALD model-only mice, the survival rate for the ALD mice that received RSV-alcohol treatment was only 50% (Figure 1B). The liver-to-body weight ratio was significantly increased by 15.9% (Figure 1D). Moreover, their serum levels of ALT and AST were significantly higher than those in the ALD model-only mice, while that of TP was significantly lower (Figure 1E). H&E staining showed the liver injury was more apparent, with lipid droplets forming around the central vein and being significantly increased by 64.3% (Figure 1F). The results showed that the RSV-alcohol treatment exacerbated ALD in mice. 

Compared with the ALD model-only mice, the ALD mice that received RSV-water treatment showed improved liver function as determined from the serum biochemical tests, indicating that RSV has some preventive and therapeutic effects on ALD (Figure 1E). However, there were significant differences in the outcomes between RSV-alcohol and RSV-water treatment on ALD. As shown in Figure 1G, the levels of ALT (*p* = 0.005), AST (*p* = 0.010), and the liver to body weight ratio (*p* = 0.001) were significantly less in RSV-water treated ALD mice than those in RSV-alcohol-treated ALD mice, while the level of TP (*p* = 0.036) was significantly higher in the former. In addition, the hepatocytes in the RSV-alcohol-treated mice were more disordered than those in the RSV-water-treated mice, with 71.9% more fat droplets forming around the central vein (*p* = 0.001) (Figure 1F, Table 2). The results demonstrate that RSV dissolved in alcohol had significantly reduced or no efficacy against ALD in mice.

### 3.3. Increased Cytotoxicity with RSV-Alcohol Combination Treatment

Compared with the vehicle treatment, neither alcohol (10 mM) nor RSV (10 μM) treatment alone affected the morphology of HL7702 cells for 24 h. However, when alcohol (10 mM) was used together with RSV (10 μM), the morphology of HL7702 cells clearly changed in a large area under the microscope (Figure 2A). The IC50 of alcohol and RSV for cell viability in the absence or presence of each other was further detected by MTT assays. The IC50 of the RSV was reduced by 20.3% in the presence of alcohol (10 mM) (Figure 2B). Consistently, the IC50 of alcohol was reduced by 17.4% in the presence of RSV (10 μM). These results suggest that RSV and alcohol may enhance the cytotoxicity of each other. 

### 3.4. Pharmacokinetic Interaction between RSV and Alcohol

The pharmacokinetic interaction between RSV and alcohol could be another reason accounting for the exacerbated liver injury by RSV-alcohol in ALD mice. Therefore, we characterised the pharmacokinetics of alcohol and RSV in the absence or presence of each other (Figure 3A). The plasma levels of alcohol, RSV, and metabolites were quantitated using HPLC-MS/MS. The typical MRM chromatograms for alcohol and *n*-butyl alcohol (IS) are shown in Figure 3B. When co-administrated with RSV (15 mg/kg), the C_max_, AUC_(0–12h)_, and t_1/2_ of alcohol (45%, *v*/*v*) were significantly increased by 52.1% (*p* = 0.004), 126.0% (*p* = 0.004), and 257.0% (*p* = 0.046), respectively, indicating an increased exposure and a reduced excretion for alcohol. The plasma concentration of alcohol in the mice that received RSV-alcohol was as high as 74.9 mmol/mL, which was significantly higher than the C_max_ of alcohol (49.3 mmol/mL, *p* = 0.003) in the mice that received alcohol alone (Figure 3C, Table 3). The data suggest that increased alcohol exposure might be a critical cause of the increased mortality and liver injury in ALD mice that received RSV-alcohol. 

The concomitant administration of RSV and alcohol not only significantly changed the pharmacokinetics of alcohol, but it also did so for RSV. The typical MRM chromatograms for RSV, its two major metabolites R3G and R3S and diethylstilbestrol (IS) are shown in Figure 3D. Compared with the mice that received RSV alone, the C_max_, AUC_(0–12h)_, and T_max_ of RSV in the mice that received RSV-alcohol were significantly reduced by 90.9% (*p* = 0.006), 95.0% (*p* = 0.012), and 56.5% (*p* = 0.002) respectively, while the t_1/2_ was significantly prolonged by 149.0% (*p* = 0.009). The pharmacokinetic parameters of R3G and R3S also changed significantly. The C_max_, AUC_(0–12h)_, and T_max_ of R3G were significantly reduced by 46.4% (*p* = 0.087), 49.1% (*p* = 0.049) and 66.7% (*p* = 0.003), respectively. In addition, t_1/2_ was significantly prolonged by 15.04% (*p* = 0.018). Similarly, the C_max_, AUC_(0–12h)_, and T_max_ of R3S were significantly decreased by 58.4% (*p* = 0.034), 56.0% (*p* = 0.012), and 52.5% (*p* = 0.011), respectively, with the t_1/2_ being prolonged by 173.0% (*p* = 0.007) (Figure 3E and Table 4). These results suggest that there was a significant pharmacokinetic interaction between RSV and alcohol.

### 3.5. Activity and Expression of Alcohol- and RSV-Metabolising Enzymes in ALD Mice

We investigated whether the pharmacokinetic interaction between alcohol and RSV could be explained by any alteration in their hepatic metabolising enzymes in ALD mice. Alcohol is mainly metabolised to acetaldehyde in the liver through Adh and Cyp2e1, and then to acetic acid through Aldh (Figure 4A). RSV is mainly transformed to R3G and R3S by Ugt1a1 and Sult1a1 in the liver, respectively (Figure 4B). In the current study, the activity of Cyp2e1 was evaluated by the production of 6-hydroxy-chlorazoxazone from the probe drug chlorazoxazone using HPLC-MS/MS (Figure 4C). Compared with those in the control group, the enzymatic activity and expression of Adh in the ALD group were significantly inhibited by 36.3% (*p* = 0.000) and 69.6% (*p* = 0.023), respectively. However, the activity and expression of Aldh were significantly increased by 114.0% (*p* = 0.000) and 57.6%, respectively. There were no significant differences in the activity or expression of Cyp2e1, Sult1a1, and Ugt1a1 between the control and ALD mice (Figure 4D,E and Table 5).

We further analysed the effects of RSV-alcohol and RSV alone on the activities and expression of the metabolising enzymes. The enzymatic activities of Adh and Cyp2e1 in the ALD mice receiving RSV-alcohol were significantly increased by 25.1% (*p* = 0.046) and 47.1% (*p* = 0.001), respectively, compared with those receiving vehicle treatment, contrasting the enzymatic activity of Aldh being decreased by 26.6% in the former (*p* = 0.070) (Figure 4E, Table 5). Consistent differences in the protein expression of Adh and Aldh were found between the two groups of mice. Interestingly, in contrast to the increased activity of Cyp2e1, the protein expression was decreased by 24.0% in the ALD mice that received RSV-alcohol compared with those that received vehicle treatment (Figure 4F, Table 5). It seemed that the co-administration of RSV with alcohol altered the effects of RSV alone on the activities and expression of those metabolising enzymes. The expression of Cyp2e1 (*p* = 0.006) and Adh (*p* = 0.083) in ALD that mice received the RSV-water treatment was significantly less than those receiving the RSV-alcohol treatment, while the activity of Aldh was higher by the RSV-water treatment (Figure 4F). Neither the RSV-water treatment nor RSV-alcohol treatment resulted in any significant changes in the expression of Sult1a1 and Ugt1a1 in ALD mice (Figure 4F, Table 5). The production of RSV metabolites was similar between the two treatments, indicating that RSV metabolism was not affected by the co-administration of alcohol (Figure 4G). Overall, these results, together with those of pharmacokinetics, suggest that the co-administration of alcohol might inhibit the absorption of RSV.

## 4. Discussion

In Eastern countries, people often drink unsupervised homemade herbal wine for disease treatment or health; however, the safety and effectiveness of this habit have not been scientifically addressed. As a nutraceutical, RSV has exhibited promising therapeutic effects against a variety of diseases, including ALD [29]. In the present study, we investigated the interaction between RSV and alcohol in an ALD mouse model as an exploratory effort to understand the outcomes of herbal wine habits. 

Similar to many natural herbs with antioxidant properties, RSV is relatively safe in clinical practice when used alone. Consistently, we found that RSV treatment alone had therapeutic benefits in ALD mice. These findings are consistent with the therapeutic effects of RSV against ALD; this has previously been reported [29,30]. Specifically, Ma et al. reported that, when administrated to rats separately from alcohol intake, RSV treatment could significantly reduce the elevation of serum ALT and AST, the accumulation of hepatic lipid droplets and TG content [31]. Moreover, Bujanda et al. demonstrated that RSV administered alone could prevent ethanol-induced liver injury, mortality and oxidative stress in mice [30]. In addition, RSV, when administrated alone, has been reported to alleviate alcohol-induced fatty liver disease in male mice by increasing the activity of the AMPK and SIRT1 pathways [32]. In these studies, RSV treatment alone was also found to prevent ethanol-induced lipid peroxidation. In the present study, surprisingly, the combined administration of RSV and alcohol increased the toxicity of alcoholic liver injury, reducing the therapeutic effects of RSV on ALD mice. This seems to contradict the therapeutic effects of RSV treatment alone, as previously reported [29,30]. However, the detrimental effects resulting from the co-administration of RSV and alcohol are also consistent with several previous findings. For example, Bujanda et al. reported moderate liver damage in mice when they received the cotreatment of alcohol and RSV [30]. Chen et al. also reported an increase in the plasma levels of TG and TC in mice resulting from such a co-administration [33]. Because the co-administration of RSV and alcohol may simulate the habit of herbal wine use, our findings have suggested an unexpected adverse effect in the use of herbal wine.

Previous studies have not provided an explanation for why the co-administration of RSV with alcohol can exacerbate alcoholic liver injury with reduced therapeutic effects. In the current study, we explored the potential mechanisms. First, we found that there was a pharmacokinetic interaction between RSV and alcohol. RSV treatment could significantly increase exposure to alcohol. Conversely, alcohol intake significantly decreased exposure to RSV. The pharmacokinetic interaction may account for, at least partially, the exacerbated liver injury caused by the co-administration of RSV and alcohol in ALD mice. Indeed, alterations in metabolising enzymes are one of the critical factors affecting pharmacokinetics [34]. We then investigated whether the pharmacokinetic interaction between RSV and alcohol was because of any alterations in the activities of the metabolising enzymes for RSV and alcohol under their concomitant administration. Adh and Cyp2e1 metabolise alcohol into acetaldehyde, which is further metabolised by Aldh to acetic acid [35] (Figure 5A). As an intermediate metabolite, acetaldehyde is even more toxic than alcohol. The accumulation of acetaldehyde often leads to the formation of adducts with various proteins and DNA, altering protein functions and promoting glutathione consumption, lipid peroxidation, and mitochondrial damage [36]. In the present study, the activity of Adh was decreased, while that of Aldh was increased in ALD mice when compared with normal mice. The results suggest that there could be reduced alcohol metabolism and the accumulation of acetaldehyde in the liver of ALD mice. RSV treatment alone showed benefits in ALD mice. However, the activity of Adh was increased, and that of Aldh decreased in ALD mice that received the RSV-alcohol combination treatment compared with those that received RSV treatment alone. The alteration in the activities of alcohol-metabolizing enzymes may lead to a higher exposure of alcohol and accumulation of acetaldehyde in the liver of ALD mice that received the combination treatment, resulting in more severe hepatic cell and tissue damage. In addition, the activities of Cyp2e1 were significantly increased in those groups as compared with normal mice, especially in the RSV-alcohol group. In ALD, an increased amount of reactive oxygen species (ROS) is produced from alcohol through Cyp2e1, causing hepatic steatosis and oxidative stress injury [37]. Although previous studies have shown that RSV treatment can inhibit the activity of Cyp2e1 and reduce liver damage [38], our results suggest that Cyp2e1 activity induced by alcohol co-administration may remove the inhibitory effect of RSV on Cyp2e1 in the RSV-alcohol combination treatment group. Thus, the increased activity of Cyp2e1 in the RSV-alcohol group might be another reason for the aggravated liver injury. In addition, we studied whether the effect of alcohol intake on the pharmacokinetics of RSV could be explained by any changes in RSV-metabolising enzymes. Ugt1a1 and Sult1a1 are the enzymes that are responsible for the formation of R3G and R3S from RSV, respectively (Figure 5B). Exposure to RSV and the two main metabolites were significantly reduced in the co-administration group compared with the RSV alone group. However, there were no significant differences in the ratio of RSV to either of the two metabolites, that is, the conversion of RSV to the two metabolites, between the two groups. These results indicate that there was no change in the activity of RSV-metabolising enzymes, which could also be reflected in their protein expression (Figure 4F). Although a future study is warranted, simultaneous alcohol intake may inhibit the absorption of RSV and reduce its therapeutic effect.

The disulfiram-ethanol reaction (DER) refers to the unpleasant symptoms associated with the consumption of either food cooked in alcohol or alcohol-based sauces and the excessive use of alcohol-containing cosmetics following the ingestion of disulfiram. DER can occur even after drinking small amounts of alcohol and even inhaling alcohol vapour from a hand sanitiser. In fact, DER causes unpleasant symptoms such as flushing, headaches and pulsating headaches [39]. Some case reports of sudden death associated with disulfiram overdose have been associated with alcohol consumption [40,41]. The habit of herbal wine intake could easily lead to DER because alcohol metabolism may be inhibited by certain components in herbal wine. In the present study, the deaths of mice in the RSV-alcohol group might be related to DER; when co-administered, RSV might significantly inhibit the metabolism of alcohol, prolong the t_1/2_ of alcohol, and aggregate liver injury because of higher exposure to alcohol.

There are limitations to our study. First, herbal wine usually contains various ingredients extracted from herbs. Exposure to alcohol in the body may result from its interaction with multiple components and is relatively complex. However, as a representative herbal ingredient, only RSV was studied in the present study, which may not reflect the overall effects of herbal medicine on alcohol metabolism in the body. Second, the pharmacokinetics of acetaldehyde were not characterised. We were unable to detect acetaldehyde in the mouse blood samples. This might be because of the high volatility of acetaldehyde and/or the low sensitivity of the analytical method. Third, although the mechanism of interaction between RSV and alcohol was found to be related to certain changes in the metabolising enzymes, the regulatory molecules need to be further defined. Finally, in addition to host factors, the gut microbiota is currently considered to be an important factor affecting drug metabolism. Further research is needed to determine whether gut microbiota is involved in the interaction between RSV and alcohol.

Because herbal wine contains a variety of substances, it is almost inevitable that there would be an interaction between the herbs and alcohol. Our findings indicated that unsupervised herbal wine intake could cause adverse effects. As for RSV, even though it has therapeutic effects in its treatment of ALD, its co-administration with alcohol may instead exacerbate the disease. For the treatment of ALD, natural products are commonly used in clinics. To achieve a better therapeutic effect and avoid aggravating alcohol damage to the body, patients should abstain from alcohol while taking these natural products.

## 5. Conclusions

The co-administration of RSV and alcohol aggravated the toxicity of ethanol to the liver while reducing the therapeutic effects of RSV, which may be related to the pharmacokinetic interactions between RSV and alcohol. Unsupervised herbal wine use is risky, and herbal wine users should be cautious about potential adverse effects.

## Figures and Tables

**Figure 1 nutrients-14-04752-f001:**
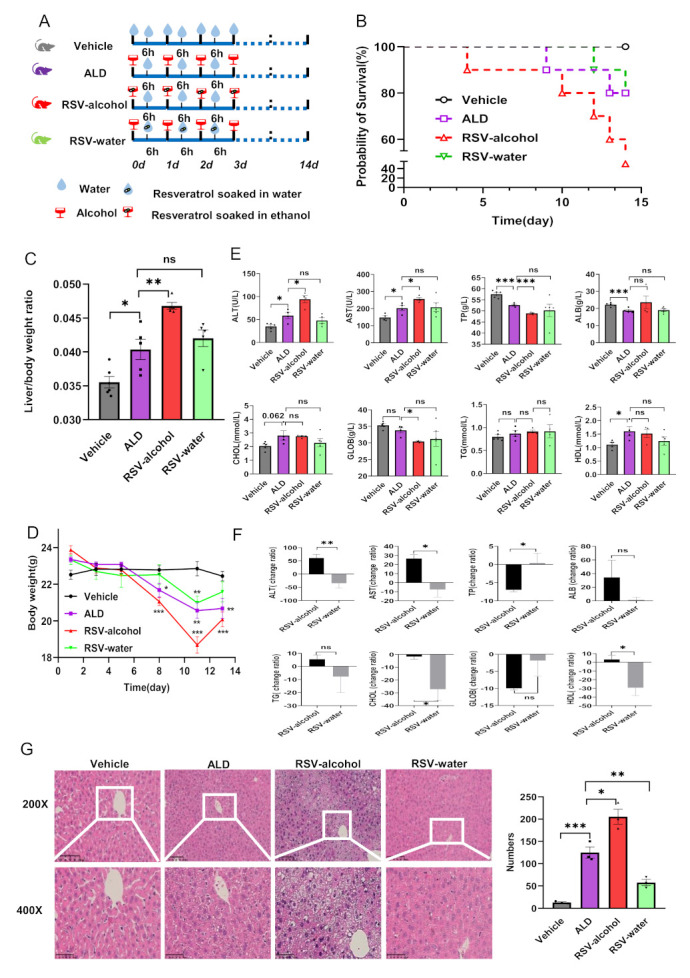
The RSV-alcohol combination worsened ALD in mice and reduced the efficacy of RSV. (**A**) Model ALD mice and mimicking the habit of herbal wine used to treat RSV. (**B**) Comparison of the survival rate. (**C**) Trends in body weight. (**D**) Comparison of the liver to body weight ratio. (**E**) Comparison of the liver function indexes. (**F**) Liver histopathology by H&E staining and lipid droplet number count. (**G**) Percentage change of liver function indexes relative to that of ALD mice. * *p* < 0.05; ** *p* < 0.01; *** *p* < 0.001; ns, no significant. The circle (○), square (□), triangle (△), and inverted triangle (▽) in the figures represent data points of different groups.

**Figure 2 nutrients-14-04752-f002:**
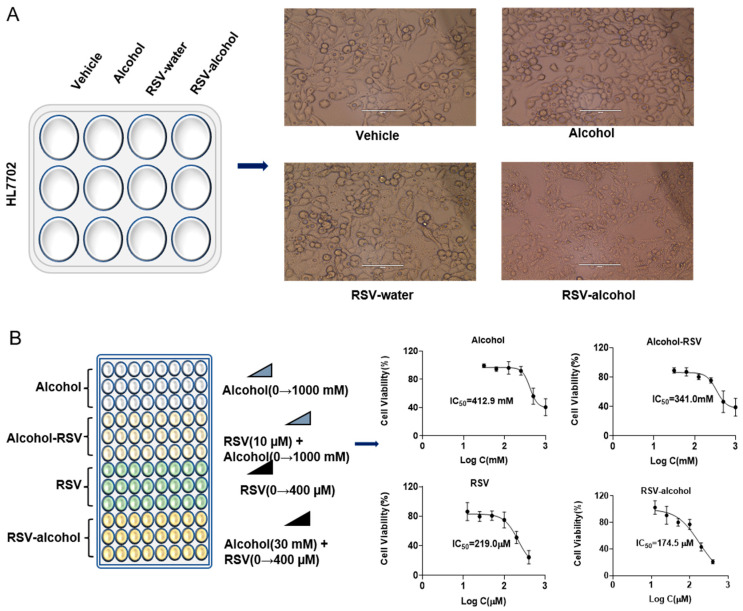
The RSV-alcohol combination worsened ALD in mice and reduced the efficacy of RSV. (**A**) RSV-alcohol combination affected the morphology of HL7702 cells. (**B**) The combination reduced the IC50s of alcohol and RSV compared with use alone.

**Figure 3 nutrients-14-04752-f003:**
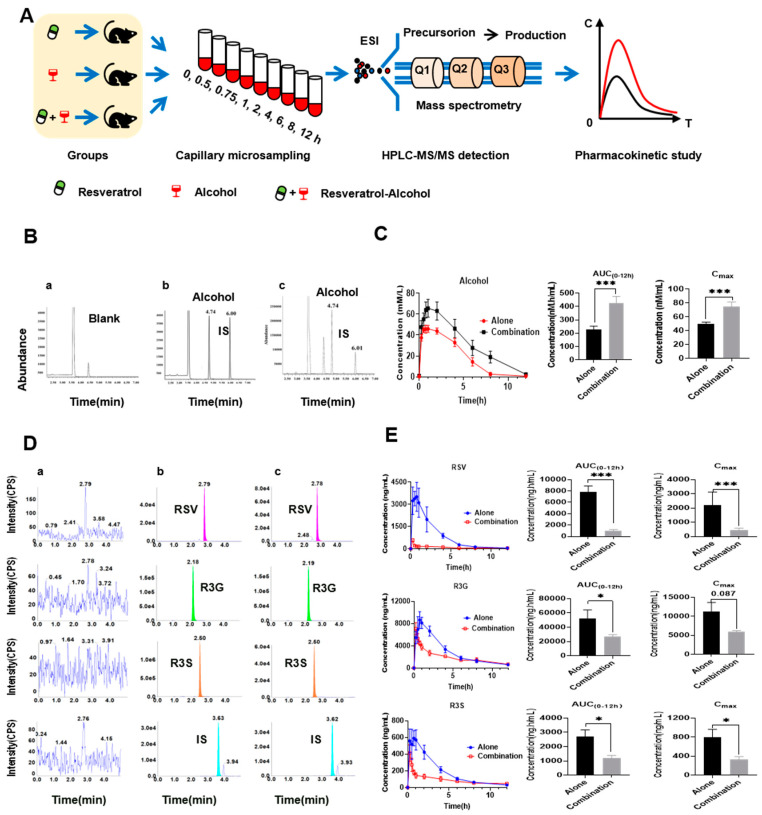
Pharmacokinetic interaction between RSV and alcohol. (**A**) Pharmacokinetic study design of RSV, alcohol and their combination in mice; (**B**) Typical multiple–reaction monitoring chromatograms of blank mouse blood(a), a blank mouse blood sample spiked with alcohol and IS(b), and a blood sample after the intragastric administration of alcohol (c). (**C**) Pharmacokinetics of alcohol in the alone and combination groups. (**D**)Typical multiple-reaction monitoring chromatograms of blank mouse blood(a), a blank mouse blood sample spiked with RSV, R3G, R3S, and IS(b), and a blood sample after the intragastric administration of RSV(c). (**E**) Pharmacokinetics of RSV and its main metabolites in the alone and combination groups. * *p* < 0.05; *** *p* < 0.001.

**Figure 4 nutrients-14-04752-f004:**
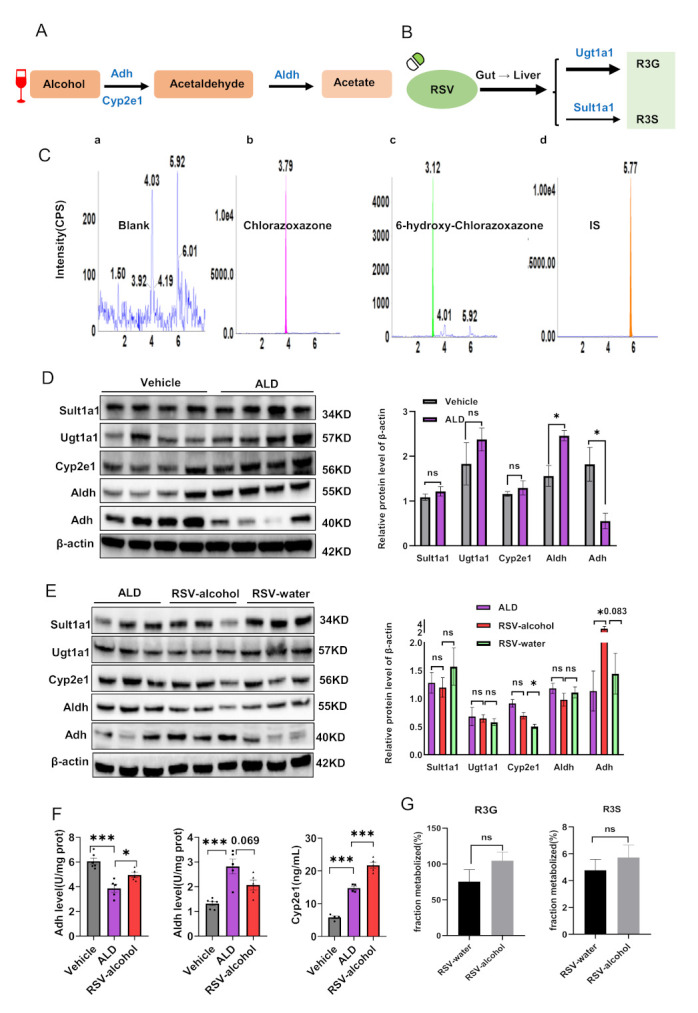
The activity and expression of alcohol- and RSV-metabolising enzymes in different treatments. (**A**) Main metabolic enzymes in the alcohol metabolic pathway. (**B**) Main metabolic enzymes in the RSV metabolic pathway. (**C**) Typical multiple-reaction monitoring chromatograms of blank liver microsomes (a), chlorazoxazone (b), 6-hydroxy-chlorazoxazone (c) and IS (d). (**D**) Metabolism enzyme expressions of RSV and alcohol in ALD mice compared with normal mice. (**E**) Comparison of metabolism enzyme activities of alcohol among vehicle, ALD, and RSV-alcohol co-administration mice. (**F**) Metabolism enzyme expressions of resveratrol and alcohol in RSV-alcohol and RSV-water treatment mice compared with ALD mice. (**G**) Comparison of RSV drug enzyme activities between the RSV-water and RSV-alcohol treatment. * *p* < 0.05; *** *p* < 0.001; ns, no significant.

**Figure 5 nutrients-14-04752-f005:**
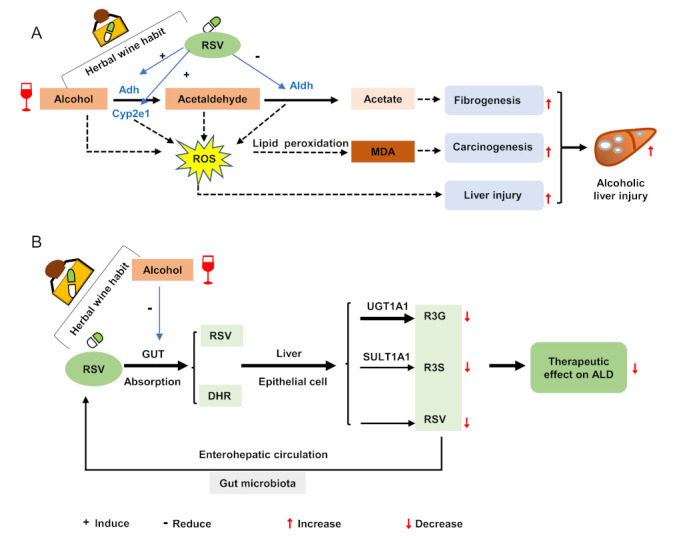
Influence of RSV−alcohol interaction when drinking herbal wine habit regularly. (**A**) Resveratrol in liquor exacerbates alcoholic liver injury through herbal wine habit administration. (**B**) Alcohol reduces the therapeutic effect of resveratrol on ALD through herbal wine habit administration. + : induce; - : reduce;↑:increase;↓: decrease.

**Table 1 nutrients-14-04752-t001:** Comparison of serum biochemical indexes among the group.

Parameters (Unit)	Groups			
	Vehicle Control	ALD Model	RSV-Alcohol	RSV-Water
ALT (U/L)	34.8 ± 37.0	58.6 ± 6.9 *↑	94.3 ± 7.8 ^#^↑	47.8 ± 6.5
AST (U/L)	148.5 ± 10.1	202.5 ± 15.3 *↑	233.8 ± 24.8 ^#^↑	208.2 ± 24.7
TP (g/L)	57.5 ± 0.7	52.6 ± 0.4 ***↓	48.9 ± 0.3 ^###^↓	50.2 ± 2.8
ALB (g/L)	22.2 ± 0.5	18.8 ± 0.7 ***↓	23.7 ± 3.7	19.0 ± 0.8
TG (mmol/L)	0.8 ± 0.0	0.9 ± 0.1	0.9 ± 0.0	1.0 ± 0.1
CHOL (mmol/L)	2.1 ± 0.1	2.8 ± 0.4	2.8 ± 0.1	2.3 ± 0.3
HDL (mmol/L)	1.1 ± 0.1	1.6 ± 0.2 *↑	1.5 ± 0.2	1.2 ± 0.2
GLOB (g/L)	35.3 ± 0.5	33.8 ± 0.8	30.4 ± 0.2 ^#^↓	31.2 ± 2.3

ALT, alanine aminotransferase; AST, aspartate aminotransferase; TP, total protein; ALB, albumin; CHOL, cholesterol; HDL, high-density lipoprotein; GLOB, globulin; RSV, resveratrol; ALD, alcoholic liver disease; RSV-alcohol, resveratrol soaked in alcohol; RSV-water, resveratrol soaked in water. Data are expressed as the mean ± SEM.* *p* < 0.05; *** *p* < 0.001, vs.Vehicle group. # *p* < 0.05; ### *p* < 0.0011, vs. ALD group. ↑: significantly increased; ↓: significantly decreased.

**Table 2 nutrients-14-04752-t002:** Percentage change in serum biochemical indexes relative to ALD mice.

Parameters (Unit)	Groups	
	RSV-Alcohol (%)	RSV-Water (%)
ALT (U/L)	61.0 ± 13.3	−34.8 ± 18.4 **
AST (U/L)	26.7 ± 4.2	−7.5 ± 8.3 *
TP (g/L)	−7.0 ± 1.0	0.4 ± 2.7 *
ALB (g/L)	34.4 ± 25.1	1.3 ± 4.2
CHOL (mmol/L)	−2.0 ± 0.5	−27.1 ± 8.7 *
TG (mmol/L)	5.5 ± 3.3	6.1 ± 16.3
HDL (mmol/L)	3.3 ± 4.7	−29.2 ± 9.0 *

ALT, alanine aminotransferase; AST, aspartate aminotransferase; TP, total protein; ALB, albumin; CHOL, cholesterol; TG, triglyceride; HDL, high-density lipoprotein; RSV-alcohol, resveratrol soaked in alcohol; RSV-water, resveratrol soaked in water. Data are expressed as the mean ± SEM. * *p* < 0.05; ** *p* < 0.01.

**Table 3 nutrients-14-04752-t003:** Comparison of pharmacokinetic parameters of alcohol administered alone and combined with resveratrol.

Parameters (Unit)	Groups	
	Alone	Combination
AUC_(0–12h)_ (nmol.h/mL)	228.3 ± 24.6	427.9 ± 47.2 **↑
AUC_(0–∞)_ (nmol.h/mL)	228.6 ± 24.6	516.7 ± 81.0 **↑
C_max_ (nmol/mL)	49.3 ± 3.1	74.9 ± 6.4 **↑
T_max_ (h)	1.0 ± 0.2	1.5 ± 0.3
t_1/2_ (h)	0.8 ± 0.2	2.9 ± 1.0

AUC_(0–12h)_, area under the plasma concentration–time curve from 0 to 12 h; AUC_(0–∞)_, area under the plasma concentration–time curve from 0 to infinity hours; C_max_, maximal concentration; T_max_, time to peak concentration; t_1/2_, eliminate half-life. Data are expressed as the mean ± SEM. ** *p* < 0.01. ↑: significantly increased.

**Table 4 nutrients-14-04752-t004:** Comparison of pharmacokinetic parameters of resveratrol and its main metabolites administered alone and combined with alcohol.

Parameters (Unit)	Resveratrol		R3G		R3S	
	Alone	Combination	Alone	Combination	Alone	Combination
AUC_(0–12h)_ (nmol.h/mL)	7779.5 ± 1103.0	1036.0 ± 181.4 ***↓	52,550.5 ± 11,692.2	26,734.7 ± 3102.9 *↓	2716.0 ± 455.0	1195.0 ± 193.7 *↓
AUC_(0-∞)_ (nmol.h/mL)	10,302.1 ± 2484.6	2215.5 ± 908.7 *↓	56,446.3 ± 11,852.6	31,948.1 ± 1964.0	2358.4 ± 390.4	1764.8 ± 363.9
C_max_ (nmol/mL)	4986.0 ± 865.0	453.6 ± 134.5 ***↓	11,211.7 ± 2441.7	6012.0 ± 259.0	801.8 ± 160.6	333.8 ± 57.2 *↓
T_max_ (h)	0.7 ±0.1	0.3 ± 0.1 ***↓	0.8 ± 0.1	0.3± 0.0 ***↓	0.7 ± 0.1	0.3 ± 0.0 *↓
t_1/2_ (h)	1.7 ± 0.2	4.2 ± 0.8 *↑	2.1 ± 0.5	5.2 ± 1.1 *↑	2.4 ± 0.3	6.5 ± 1.4 ***↑

AUC_(0–12h)_, area under the plasma concentration-time curve from 0 to 12 h; AUC_(0–∞)_, area under the plasma concentration-time curve from 0 to infinity hours; C_max_, maximal concentration; T_max_, time to peak concentration; t_1/2_, eliminate half-life. R3G, trans-resveratrol-3-o-β-glucuronide; R3S, trans-resveratrol-3-sulfate salt. Data are expressed as the mean ± SEM. * *p* < 0.05; *** *p* < 0.001. ↑: significantly increased; ↓: significantly decreased.

**Table 5 nutrients-14-04752-t005:** Comparison of metabolizing enzyme activities for alcohol and resveratrol among the groups.

Parameters (Unit)	Metabolic Compounds		Enzyme Activities	
		Vehicle	ALD Model	RSV-Alcohol
Adh (U/mg prot)	alcohol	6.1 ± 0.3	3.9 ± 0.3 ***↓	4.8 ± 0.3 ^#^↑
Aldh (U/mg prot)	alcohol	1.3 ± 0.1	2.8 ± 0.3 ***↑	2.1 ± 0.2
Cyp2e1 (ng/mL)	alcohol	5.9 ± 0.4	14.8 ± 0.6 ***↑	21.7 ± 0.9 ^###^↑
Ugt1a1 (%)	RSV	75.4 ± 16.8	89.0 ± 10.3	/
Sult1a1 (%)	RSV	4.8 ± 0.8	4.9 ± 0.8	/

Adh, alcohol dehydrogenase; Aldh, aldehyde dehydrogenase; Cyp2e1, cytochrome P4502e1; Ugt1a1, Uridine diphosphate glucuronic acid transferase 1a1; Sult1a1, sulfatase transferase 1a1; ALD, alcoholic liver disease; RSV-alcohol, resveratrol soaked in alcohol. Data are expressed as the mean ± SEM. /, undetected. *** *p* < 0.001, vs. Vehicle group. # *p* < 0.05; ### *p* < 0.001, vs. ALD group. ↑: significantly increased; ↓: significantly decreased.

## Data Availability

Data is contained within the article.

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
