# Peer review of "Resveratrol in Liquor Exacerbates Alcoholic Liver Injury with a Reduced Therapeutic Effect in Mice: An Unsupervised Herbal Wine Habit Is Risky"

_nutrients, 2022, doi:10.3390/nu14224752_

Round 1

Reviewer 1 Report

In this paper Zhang et al. investigated the effect of Resveratrol in ethanol against alcoholic liver disease in mice. Resveratrol (15mg/kg) soaked in 45% ethanol (Resveratrol-alcohol) was administrated via gavage to the mice with alcohol consumption-induced ALD. Resveratrol soaked in water was the treatment control. Authors reported that compared to Resveratrol-water group, a higher rate of mortality was found in Resveratrol-alcohol group. Resveratrol significantly increased the exposure of alcohol by 126.0%, accompanied by a significantly inhibition on the ethanol metabolic pathway. In contrast, alcohol consumption significantly reduced the exposure of Resveratrol by 95.0%.

Interesting data highlighting pharmakinetic aspects on the interaction of Resveratrol with alcohol leading to the alcoholic liver injury with reduced therapeutic effects.

I have only minor issues:

1.      Fine English editing

2.      Can author explain the increase in HDL in  ALD model?

Author Response

We have carefully read your comments and have tried our best to address them one by one. Enclosed are our replies, which we hope you’ll find satisfactory. The modifications were highlighted by yellow in the revised manuscript.

Editor :

Point 1: Manuscripts containing original descriptions of research conducted on humans or experimental animals must contain details of approval by aproperly constituted research ethics committee. Besides, the project identification code, date of approval and name of the ethics committee or institutional review board should be mentioned in the Methods section. Please also add your manuscript.

Response 1: Thank you very much for this valuable proposal! In response to this proposal, we have described the ethical details of animal experiments in the “Materials and Methods” of the manuscript (Line 104-106), and added the ethical project identification code, approval date and the name of the ethics committee. At the same time, we also mentioned these contents in the "Statement of the Institutional Review Committee" (Line 570-571). These additions have been marked in red in the corresponding positions of the manuscript.

Revision details:the Administration Committee of Experimental Animals of Central South University (No:2018sydw0219, Date:  Oct,18th 2018, Changsha, Hunan, China).

Reviewer #1:

In this paper Zhang et al. investigated the effect of Resveratrol in ethanol against alcoholic liver disease in mice. Resveratrol (15mg/kg) soaked in 45% ethanol (Resveratrol-alcohol) was administrated via gavage to the mice with alcohol consumption-induced ALD. Resveratrol soaked in water was the treatment control. Authors reported that compared to Resveratrol-water group, a higher rate of mortality was found in Resveratrol-alcohol group. Resveratrol significantly increased the exposure of alcohol by 126.0%, accompanied by a significantly inhibition on the ethanol metabolic pathway. In contrast, alcohol consumption significantly reduced the exposure of Resveratrol by 95.0%.

Interesting data highlighting pharmakinetic aspects on the interaction of Resveratrol with alcohol leading to the alcoholic liver injury with reduced therapeutic effects.

I have only minor issues:

 Point 1: Fine English editing

Response 1: The manuscript has been proofread and revised by native speakers who are knowledgeable in this field. We hope the language quality now meets  the standards of the journal.

 Point 2: Can author explain the increase in HDL in ALD model?

 Response 2: This is an interesting question; we may not give you a satisfactory answer from the present study, though it has inspired us to pursue another interesting topic in future study. A high level of HDL could be a good thing for health, while alcohol as a toxin could increase its level, but this does not mean that alcohol is a good thing. However, it does exist in our experiment. Based on your question, we also looked at the literature. Previous studies also reported that moderate alcohol consumption will affect lipoprotein metabolism and is related to an increase of high-density lipo-protein cholesterol (HDL-C), which has been observed in multiple epidemiological and Mendelian randomised studies [1,2]. The studies explained that alcohol can increase the production of apolipoprotein A-I (apoAI) in the liver, which is the main protein component of high-density lip-oprotein granules [3]. Many thanks to you for providing us with further research directions.

References:

[1] Tolstrup, J. S., Grønbek, M. & Nordestgaard, B. G. Alcohol Intake, Myocardial Infarction, Biochemical Risk Factors, and Alcohol Dehydrogenase Genotypes. Circ Cardiovasc Genet 2, 507–514 (2009).

[2] Tabara, Y. et al. The causal effects of alcohol on lipoprotein subfraction and triglyceride levels using a Mendelian randomization analysis: The Nagahama study. Atherosclerosis 257, 22–28 (2017)

[3] Rimm, E. B., Williams, P., Fosher, K., Criqui, M. & Stampfer, M. J. Moderate alcohol intake and lower risk of coronary heart disease: meta-analysis of effects on lipids and haemostatic factors. BMJ 319, 1523–1528 (1999).

Reviewer #2:

The study of Song-xia Zhang proves the negative effect of the combination of reservatol with alcohol in an ALD animal model. It is a well designed study and offers a lot of results towards this direction. The included experiments include suficient experimental groups and the obtained results confirm the hypothesis in the present study.

 Point 1: There is a small typo in Line 17: "...plays a vital role" (not an vital role)

Response 1:

Thank you for pointing out this typographical error. The revised content has been revised in the corresponding position of the manuscript (Line 73) and marked in red.

In addition, the manuscript has been furthered by native speakers to meet the high-quality standards of the journal.

 Point 2: The main concern that authors need to explain is whether the level of alcohol differs between the groups ALD y RSV-alcohol. If authors cannot ensure that, this fact may be the reason that explains the observed differences.

Response 2:

This is a very good question, which will help us further improve the quality of the manuscript. In fact, alcohol use doses were the same in the ALD and RSV–alcohol groups; the difference is that in the RSV–alcohol group, RSV was administrated together with the alcohol used in the ALD group, but not with an extra amount of alcohol. However, there was an incorrect expression in our abstract that may have misled readers. RSV dissolved in a fixed 45% of the alcohol was not right because the alcohol concentration used in the ALD model was different in two stages. In the first week of the experiment, the mice in the ALD group were fed with 30.3% alcohol by gavage. In the second week of modelling, the mice in the ALD group used 50.6% alcohol for modelling. Thus, the RSV together used in the RSV–alcohol groups should be 30.3% or 50.6% alcohol, respectively. To avoid this possible confusion, we deleted the part about 45% of the alcohol from the abstract.

Best Regards

Yours Sincerely

Yao Chen

Department of Clinical Pharmacology

Xiangya Hospital, Central South University

Changsha, Hunan 410008, China.

E-mail: cbohua@csu.edu.cn

Reviewer 2 Report

The study of Song-xia Zhang proves the negative effect of the combination of reservatol with alcohol in an ALD animal model. It is a well designed study and offers a lot of results towards this direction. The included experiments include suficient experimental groups and the obtained results confirm the hypothesis in the present study.

There is a small typo in Line 17: "...plays a vital role" (not an vital role)

The main concern that authors need to explain is whether the level of alcohol differs between the groups ALD y RSV-alcohol. If authors cannot ensure that, this fact may be the reason that explains the observed differences.

Author Response

We have carefully read your comments and have tried our best to address them one by one. Enclosed are our replies, which we hope you’ll find satisfactory. The modifications were highlighted by yellow in the revised manuscript.

Editor :

Point 1: Manuscripts containing original descriptions of research conducted on humans or experimental animals must contain details of approval by aproperly constituted research ethics committee. Besides, the project identification code, date of approval and name of the ethics committee or institutional review board should be mentioned in the Methods section. Please also add your manuscript.

Response 1: Thank you very much for this valuable proposal! In response to this proposal, we have described the ethical details of animal experiments in the “Materials and Methods” of the manuscript (Line 104-106), and added the ethical project identification code, approval date and the name of the ethics committee. At the same time, we also mentioned these contents in the "Statement of the Institutional Review Committee" (Line 570-571). These additions have been marked in red in the corresponding positions of the manuscript.

Revision details:the Administration Committee of Experimental Animals of Central South University (No:2018sydw0219, Date: Oct, 18th 2018, Changsha, Hunan, China).

Reviewer #1:

In this paper Zhang et al. investigated the effect of Resveratrol in ethanol against alcoholic liver disease in mice. Resveratrol (15mg/kg) soaked in 45% ethanol (Resveratrol-alcohol) was administrated via gavage to the mice with alcohol consumption-induced ALD. Resveratrol soaked in water was the treatment control. Authors reported that compared to Resveratrol-water group, a higher rate of mortality was found in Resveratrol-alcohol group. Resveratrol significantly increased the exposure of alcohol by 126.0%, accompanied by a significantly inhibition on the ethanol metabolic pathway. In contrast, alcohol consumption significantly reduced the exposure of Resveratrol by 95.0%.

Interesting data highlighting pharmakinetic aspects on the interaction of Resveratrol with alcohol leading to the alcoholic liver injury with reduced therapeutic effects.

I have only minor issues:

 Point 1: Fine English editing

Response 1: The manuscript has been proofread and revised by native speakers who are knowledgeable in this field. We hope the language quality now meets  the standards of the journal.

 Point 2: Can author explain the increase in HDL in ALD model?

 Response 2: This is an interesting question; we may not give you a satisfactory answer from the present study, though it has inspired us to pursue another interesting topic in future study. A high level of HDL could be a good thing for health, while alcohol as a toxin could increase its level, but this does not mean that alcohol is a good thing. However, it does exist in our experiment. Based on your question, we also looked at the literature. Previous studies also reported that moderate alcohol consumption will affect lipoprotein metabolism and is related to an increase of high-density lipo-protein cholesterol (HDL-C), which has been observed in multiple epidemiological and Mendelian randomised studies [1,2]. The studies explained that alcohol can increase the production of apolipoprotein A-I (apoAI) in the liver, which is the main protein component of high-density lip-oprotein granules [3]. Many thanks to you for providing us with further research directions.

References:

[1] Tolstrup, J. S., Grønbek, M. & Nordestgaard, B. G. Alcohol Intake, Myocardial Infarction, Biochemical Risk Factors, and Alcohol Dehydrogenase Genotypes. Circ Cardiovasc Genet 2, 507–514 (2009).

[2] Tabara, Y. et al. The causal effects of alcohol on lipoprotein subfraction and triglyceride levels using a Mendelian randomization analysis: The Nagahama study. Atherosclerosis 257, 22–28 (2017)

[3] Rimm, E. B., Williams, P., Fosher, K., Criqui, M. & Stampfer, M. J. Moderate alcohol intake and lower risk of coronary heart disease: meta-analysis of effects on lipids and haemostatic factors. BMJ 319, 1523–1528 (1999).

Reviewer #2:

The study of Song-xia Zhang proves the negative effect of the combination of reservatol with alcohol in an ALD animal model. It is a well designed study and offers a lot of results towards this direction. The included experiments include suficient experimental groups and the obtained results confirm the hypothesis in the present study.

 Point 1: There is a small typo in Line 17: "...plays a vital role" (not an vital role)

Response 1:

Thank you for pointing out this typographical error. The revised content has been revised in the corresponding position of the manuscript (Line 73) and marked in red.

In addition, the manuscript has been furthered by native speakers to meet the high-quality standards of the journal.

 Point 2: The main concern that authors need to explain is whether the level of alcohol differs between the groups ALD y RSV-alcohol. If authors cannot ensure that, this fact may be the reason that explains the observed differences.

Response 2:

This is a very good question, which will help us further improve the quality of the manuscript. In fact, alcohol use doses were the same in the ALD and RSV–alcohol groups; the difference is that in the RSV–alcohol group, RSV was administrated together with the alcohol used in the ALD group, but not with an extra amount of alcohol. However, there was an incorrect expression in our abstract that may have misled readers. RSV dissolved in a fixed 45% of the alcohol was not right because the alcohol concentration used in the ALD model was different in two stages. In the first week of the experiment, the mice in the ALD group were fed with 30.3% alcohol by gavage. In the second week of modelling, the mice in the ALD group used 50.6% alcohol for modelling. Thus, the RSV together used in the RSV–alcohol groups should be 30.3% or 50.6% alcohol, respectively. To avoid this possible confusion, we deleted the part about 45% of the alcohol from the abstract.

Best Regards

Yours Sincerely

Yao Chen

Department of Clinical Pharmacology

Xiangya Hospital, Central South University

Changsha, Hunan 410008, China.

E-mail: cbohua@csu.edu.cn
